# Co-Administration of Influenza and COVID-19 Vaccines: Policy Review and Vaccination Coverage Trends in the European Union, UK, US, and Canada between 2019 and 2023

**DOI:** 10.3390/vaccines12020216

**Published:** 2024-02-19

**Authors:** Roel C. A. Achterbergh, Ian McGovern, Mendel Haag

**Affiliations:** 1Netherlands School of Public & Occupational Health, 3527 GV Utrecht, The Netherlands; 2Center for Outcomes Research and Epidemiology, CSL Seqirus, Waltham, MA 02451, USA; ian.mcgovern@seqirus.com; 3Center for Outcomes Research and Epidemiology, CSL Seqirus, 1105 BJ Amsterdam, The Netherlands

**Keywords:** COVID-19, influenza, vaccine co-administration, vaccination policy, public health surveillance, vaccine coverage

## Abstract

Recommending co-administration of influenza and COVID-19 vaccines has emerged as a strategy to enhance vaccination coverage. This study describes the policy on co-administration and uptake of influenza and COVID-19 vaccination in Europe, the United Kingdom, the United States, and Canada between 2019 and 2023. We collected co-administration policy data from governmental websites, national health organizations, and newspapers. Influenza vaccination coverage among persons ≥65 years and COVID-19 vaccination coverage rates among persons ≥60 years or the general population were collected using national databases, the ECDC database, or ourworldindata.org between 2019 and 2023. Descriptive analyses were used. We collected data from 30/32 (94%) countries on vaccination policy in seasons 2021–2022 and 2022–2023, with most countries (25/30 to 30/30) having policies recommending co-administration. For influenza vaccination coverage, we collected data from 29/32 (91%, 2019–2020), 28/32 (88%, 2020–2021), 27/32 (84%, 2021–2022), and 6/32 (19%, 2022–2023) countries. COVID-19 vaccination was collected from 32/32 (2020–2021), 31/32 (97%, 2021–2022), and 24/32 (75%, 2022–2023) countries. Influenza vaccination coverage increased from 2019–2020 to 2021–2022. COVID-19 vaccination coverage was higher among countries with higher influenza vaccination coverage. By 2022–2023, all countries included implemented a policy supporting co-administration. A positive correlation existed between higher influenza vaccination coverage and higher COVID-19 vaccination rates.

## 1. Background

The global pandemic caused by the COVID-19 virus has had profound health, social and economic impacts worldwide [1,2,3]. Vaccination has emerged as a crucial strategy to mitigate the spread and severity of COVID-19 [3]. Sequentially, influenza remains a significant respiratory illness that poses a continual threat to global health [4]. Annual influenza vaccination is recommended, particularly for individuals at high risk of severe illness or complications from influenza and those over 65 years of age [4,5].

Ensuring high vaccination uptake against both COVID-19 and influenza is crucial to reduce the burden of these respiratory illnesses. To encourage vaccination, various strategies have been employed, including the co-administration of both vaccines simultaneously or offering vaccinations separately [6]. The effect of policy on vaccination coverage remains an area with limited research.

Vaccination hesitancy has been labeled as one of the top ten threats to global health by the World Health Organization (WHO) and has emerged as a concerning challenge, posing barriers to achieving optimal vaccination coverage [7,8]. Causes of vaccine hesitancy encompass concerns over safety, lack of trust, lack of need for vaccination, and cultural reasons [9]. Importantly, certain causes exhibit specificity to particular countries, reflecting the nuanced influences that contribute to hesitancy within diverse cultural and regional contexts [9]. In light of these factors, understanding the effect of co-administration policies on vaccination coverage in different countries is crucial for effective public health planning and intervention strategies.

Annual influenza vaccination campaigns typically span from August to March in the Northern Hemisphere. There is an emphasis on people being vaccinated during September/October, prior to the peak influenza season that generally occurs between December and February [10,11]. A previous review of influenza vaccination policy and coverage within the European Union between 2015 and 2018 showed a general consensus that older age groups should be vaccinated [5]. However, there was variability in the definition of older age, with some countries targeting individuals aged 60 years and older, while others set the threshold at 65 years and older. The United Kingdom (UK) was the only country that almost reached the WHO target of 75% among the elderly in the 2015–2016 and 2016–2017 influenza seasons. Currently, the Centers for Disease Control and Prevention (CDC) of the United States (US) recommends all persons 6 months and older get an annual influenza vaccine, including healthy adults [12].

The first COVID-19 vaccinations began between December until March 2020–2021, approximately a year after the onset of the pandemic [13]. Initial boosters were recommended by the CDC for moderately and severely immunocompromised persons in August 2021 and eventually for all persons aged ≥18 years in November 2021 [14]. COVID-19 booster campaigns did not always align with the influenza vaccination schedule. On 21 October 2021, the World Health Organization (WHO) recommended the co-administration of influenza and COVID-19 vaccines [15].

This study aims to describe the policy on the co-administration of influenza and COVID-19 vaccination in the European Union (EU), Iceland, Norway, the UK, the US, and Canada. Also, it aims to describe vaccination uptake of influenza and COVID-19 within these countries between 2019 and 2023.

The findings of this study aim to contribute to the existing body of knowledge by providing an overview of the policies surrounding the co-administration of COVID-19 and influenza vaccines and vaccination uptake. It aims to inform policymakers, researchers, and public health authorities in developing evidence-based strategies to enhance vaccination coverage. These insights will be relevant for optimizing vaccination efforts against vaccine-preventable diseases and mitigating their impact on global health for COVID-19 and beyond.

## 2. Methods

### 2.1. Study Design

In this review, we describe the policy on co-administration and vaccination coverage of COVID-19 and influenza in all EU countries (Austria, Belgium, Bulgaria, Croatia, Cyprus, Czech Republic, Denmark, Estonia, Finland, France, Germany, Greece, Hungary, Italy, Ireland, Latvia, Lithuania, Luxembourg, Malta, the Netherlands, Poland, Portugal, Romania, Spain, Slovakia, Slovenia, and Sweden), Iceland, Norway, UK, US, and Canada. We did not include countries in the Southern Hemisphere due to the temporal variation in the emergence of influenza, which occurs at a different time compared to the Northern Hemisphere [10,11]. All data were collected retrospectively and accessed in May 2023.

### 2.2. Data Sources

#### 2.2.1. Policy Data

We collected data from vaccination policies on co-administration recommendations. We divided the collected data on policy on co-administration into two categories: (1) recommending co-administration (policies recommending co-administration) and (2) not recommending co-administration (co-administration allowed but not recommended or not allowed and not recommended). Policies that changed into recommending co-administration during the influenza season were categorized as recommending co-administration. We collected data during the influenza seasons 2021–2022 and 2022–2023. We collected data from governmental websites, national health organizations, or, if otherwise unavailable, national newspapers.

#### 2.2.2. Influenza Vaccination Coverage Data

Data on influenza vaccination coverage was preferentially collected from national health organizations or governmental websites or, otherwise, if unavailable, from ‘Our World in Data’, WHO, or OECD. We only collected data among persons aged 65 years and older due to different definitions of risk groups per country and the limited availability of influenza vaccination coverage data for the general population or other risk groups [5]. We used data from the 2019–2020 influenza season (pre-pandemic), 2020–2021, 2021–2022, and, if available, 2022–2023 influenza seasons.

#### 2.2.3. COVID-19 Vaccination Coverage Data

Data on COVID-19 vaccination coverage was collected first via ‘Our World in Data’. Contrary to influenza, COVID-19 vaccine coverage was not reported specifically for 65 years and older or specified by primary or booster vaccination. Hence, basic vaccination coverage (i.e., at least one vaccination) was used, as well as the number of vaccine doses given per 100 persons in the general overall population. Coverage rates could exceed 100 vaccinations per 100 persons due to the booster dose regimens. We used data from 2021, 2022, and 2023. The same endpoint as influenza season was used—31 March—for each year. If data for 31 March was missing, we used the closest data point available. Secondly, to ensure more consistency in age groups, we also collected data from the ‘ECDC Vaccine tracker’ for countries within the EU/EEA. COVID-19 vaccine coverage was not reported specifically for 65 years and older but for 60 years and older. We reported coverage with a full primary course, a first booster, and a second booster. Again, the same endpoint as influenza season was used—31 March—for each year (i.e., week 14 in 2021 and week 13 in both 2022 and 2023). If no data were available, which included all countries outside the EU/EEA, we used data from ‘Our world in Data’ as described above.

#### 2.2.4. Informed Consent

Informed consent was not applicable. This study used exclusively national anonymized and publicly available data.

#### 2.2.5. Statistical Methods

We used descriptive analysis to summarize the data. We performed a correlation analysis using Excel to quantify the strength of the relationship between influenza and COVID-19 vaccination coverage.

To calculate the average influenza vaccination coverage, we used the average of all available countries in the 2019–2020 season as a baseline. For subsequent seasons, we calculated the absolute difference between the previous and the next season per country. For example, the influenza vaccination coverage in France was 52.0% in 2019–2020 and 59.9% in 2020–2021. The absolute difference (7.9%) was calculated for each country. We added the absolute average difference to the absolute average of the previous year. Countries with missing data on vaccination coverage were left out of the analysis; only the years with missing data and years that could be added to the data were added.

## 3. Results

### 3.1. Availability of Data

We collected data on policy on co-administration (see Appendix A Table A1) in 30 out of the 32 countries for the 2021–2022 season (missing Iceland and Luxembourg) and 30 in the 2022–2023 season (missing Latvia and Cyprus). Only five countries (Bulgaria, Cyprus, Hungary, Norway, and Slovakia) had a policy not recommending co-administration in 2021–2022. Out of these five countries, Cyprus was the only country with missing data for 2022–2023. All 30 included countries for the 2022–2023 season recommended co-administration.

Twenty-six countries had data on influenza vaccine coverage (see Appendix A Table A2) among persons ≥65 years of age for all seasons between 2019 and 2022. We were unable to collect any data on influenza vaccination coverage for two countries (Belgium and Cyprus). For four countries, we collected partial data: Austria and Poland, only the 2019–2020 season; for the Czech Republic, we missed data for the 2019–2020 season; and for Latvia, we missed the 2021–2022 season. There were six countries (Denmark, France, Ireland, Portugal, the UK, and the US) with available data for the 2022–2023 season.

Using ‘Our World in Data’, we collected data on basic vaccination coverage for COVID-19 (see Appendix A Table A3) in the general population on 31 March 2021 in 32/32 countries and in 31/32 countries in 2022 (missing Romania; see Appendix A Table A3). We collected data on 24/32 countries for 2023 (missing Iceland, Latvia, Netherlands, Norway, Romania, Slovakia, Slovenia, and the UK). For the number of vaccine doses given per 100 citizens, we collected data on all countries in 2021 and 2022 and in 24/32 countries in 2023.

We collect data via ‘ECDC Vaccine Tracker’ on basic vaccination coverage for COVID-19 (see Appendix A Table A3) among persons ≥60 years of age in 29/32 countries on 31 March 2021 (missing UK, US, and Canada). We collected data on 29/32 countries for the first booster in 2022 and 29/32 countries for the second booster in 2023 (missing the UK, the US, and Canada).

### 3.2. Vaccination Coverage

The average influenza coverage data among persons ≥65 years of age in 2019–2020 was 41.3% (see Figure 1). The absolute average coverage increased by 5.9% in 2020–2021 compared to 2019–2020. The year after, the absolute average increased again by 0.9% in 2021–2022. In 2022–2023, with data from only six countries available, the absolute average decreased again by 0.4% compared to the prior season. Only Portugal, the UK, Denmark, and Ireland reached the 75% goal set by the WHO. The US reached the 75% goal in the 2020–2021 season, but vaccination coverage decreased, reaching 71.0% in 2022–2023. Countries in Eastern Europe reached lower levels of influenza coverage compared to countries in Western and Northern Europe and North America.

Most countries that had a relatively high influenza vaccination coverage among persons ≥65 years of age also had a high coverage of ≥1 COVID-19 vaccinations in the overall population (see Figure 2a) as well as among persons ≥60 years of age (Figure 2c) at the end of the influenza season 2022. Most countries that had relatively high influenza vaccine coverage also had a relatively higher number of COVID-19 vaccinations per 100 citizens (see Figure 2b). The correlation coefficients were 0.598 for Figure 2a, 0.690 for Figure 2b, and 0.593 for Figure 2c. Most countries in Eastern Europe reached lower levels of vaccination coverage for both vaccines compared to countries in Western and Northern Europe and North America. In addition, three out of four countries not recommending co-administration in 2021–2022 were in Eastern Europe, with Norway being an exception.

Figure 3a shows the influenza vaccination coverage among persons ≥65 and the number of COVID-19 vaccinations given per 100 persons on 31 March 2021, 2022, and 2023. Figure 3b shows the influenza vaccination coverage among persons ≥65 years of age and coverage of the primary series, first booster, and second booster of COVID-19 vaccinations among persons ≥60 years of age given at week 14 in 2021 and week 13 in 2022 and 2023, respectively. Countries with a relatively high number of COVID-19 vaccinations given per 100 persons also had relatively higher influenza vaccination coverage. While the US had a relatively high COVID-19 vaccination rate in 2021 and a high influenza vaccination coverage, the number of new COVID-19 vaccines administered per capita in 2022 and 2023 was lower relative to other countries evaluated. In the last year with available data, between March 2022 and March 2023, 7 out of 24 countries increased their number of COVID-19 vaccinations by ≤10 per 100 persons (Poland, Estonia, Cyprus, Bulgaria, Czech Republic, Croatia, and Ireland). In the same period, there were also seven countries which increased the number of COVID-19 vaccinations per 100 persons by ≥25 (Sweden (46), Portugal (42), Canada (38), Belgium (37), Denmark (35), the US (32), and Finland (27)).

## 4. Discussion

The findings of this study show that most countries included in the analysis had a policy supporting the co-administration of influenza and COVID-19 vaccines. Only four countries did not recommend co-administration during the 2021–2022 season. In the subsequent season, all included countries had policies supporting co-administration. Though variations did exist between countries, influenza vaccination coverage on average increased in the years 2019–2022 among persons 65 years of age and older. Notably, we observed a positive correlation between countries with higher influenza vaccination coverage and higher COVID-19 vaccination rates.

### 4.1. Interpretation

We found that most included countries issued co-administration policies in 2021–2022 with a shift towards universal support in the subsequent season. These results highlight the general acceptance and adoption of co-administration on the country level. However, it is worth noting that despite the existence of these policies, there is limited available data on the adherence to co-administration policies.

We observed a large difference in both influenza as well as COVID-19 vaccination coverage in countries in Eastern Europe versus Northern America and the rest of Europe. These findings are in line with previous findings [5,16]. Variations in age, education, accessibility, risk perception, trust in (health) authorities, availability of healthcare professionals, and vaccination infrastructure can significantly influence vaccination uptake [9,17]. Social disparities can play a role in vaccination coverage as well [18].

The positive correlation observed between influenza vaccination coverage and COVID-19 vaccination rates indicates that countries with higher influenza vaccination coverage tended to have higher COVID-19 vaccination coverage as well. These results are consistent with previous studies at the individual level, which have shown that individuals who receive an influenza vaccination are more likely to accept COVID-19 vaccinations [19,20,21]. Moreover, this correlation aligns with the WHO’s perspective at the systemic level. The WHO emphasizes integrated pathogen planning by using similar systems and capacities to respond to different respiratory pathogens [22]. This unified approach allows for efficient preparedness, covering both known and novel pathogens [22]. Our findings suggest that countries may have benefited from leveraging existing infrastructure and the public’s acceptance of influenza vaccination, thereby equipping them with the tools to promote and achieve higher COVID-19 vaccination coverage rates.

Additionally, there were six countries providing data on influenza vaccination coverage for the 2022–2023 season. During the analyzed period, five of these six countries achieved the highest levels of influenza vaccination coverage of all included countries. This finding possibly underscores that countries that invest the most in a comprehensive and integrated approach to infectious disease prevention through vaccination also tend to be the ones that have earlier/more complete data on vaccine coverage.

We found an overall increase in influenza vaccination coverage in the seasons 2019–2020 through 2021–2022. Among the six countries with data for the 2022–2023 season, we found a small decrease (0.4%) in influenza vaccination coverage observed, which warrants further investigation. This decrease in coverage aligns with reported declines in childhood immunization programs in several countries, potentially indicating a broader issue of vaccine hesitancy [9,23,24,25,26]. Understanding the exact reasons behind this decrease is crucial to inform targeted interventions and ensure optimal vaccination coverage in future seasons. Efforts should focus on addressing barriers to vaccination and addressing public health concerns or misconceptions that may have arisen during the pandemic. Enhancing public awareness campaigns on vaccine safety and improving accessibility and trust to vaccination services, particularly among underserved populations, could contribute to higher vaccination coverage [9].

To counter vaccine hesitancy, the promotion of vaccine literacy is occasionally advocated and might, therefore, be crucial in navigating the landscape of co-administration policies. Vaccine literacy is related to health literacy and is defined as the ability of people to access, process, and understand basic vaccination knowledge and vaccination services, as well as to assess the potential consequences and risks of their behavior and make health-related decisions [27]. Beyond the realm of policymakers and healthcare workers, acknowledging the essential roles played by the media, communities, schools, and the population itself is crucial for fostering a vaccine-literate environment [28]. As noted by Larson et al., addressing vaccine literacy involves not only providing accurate information but also debunking prevalent myths and enhancing overall health literacy [29]. Effective collaboration and partnerships among all stakeholders are paramount. Misinformation and misconceptions can significantly impact public perceptions of co-administration strategies. This study reveals a noteworthy consistency in co-administration policies across nearly all countries, contributing to a clearer and more cohesive public health message.

Crucial for the optimization of vaccination strategies is the understanding of how the public comprehends co-administration policies. A study among a representative sample of Italian adults revealed that only 22.9% were favorable to vaccine co-administration, while 16.6% declared firm unwillingness [30]. The remaining 60.5% fell into the category of hesitancy. Similarly, a national, quota-based cross-sectional sample of parents in the United States demonstrated that only 10.6% were willing to have their adolescent child receive COVID-19 and routine vaccines simultaneously, and 18.5% would follow the healthcare provider’s recommendation [31]. While both studies indicate low willingness, it is noteworthy that these investigations were conducted in the early stages of the pandemic, and outcomes may currently differ. In childhood vaccination, the combination of vaccines is associated with improved coverage rates [32].

The CDC has recommended RSV vaccines for adults on 29 June 2023 [33]. However, there is currently no clear policy on the co-administration of new (respiratory) vaccines, such as RSV, alongside vaccines such as influenza and COVID-19. Co-administration is a complex decision involving considerations of safety, effectiveness, feasibility, and cost-effectiveness, among others [34]. While studies have examined the combinations of influenza and COVID-19 vaccines and co-administration is common practice for travel vaccines and national immunization programs to protect children, the interactions between existing and newly developed vaccines have not been as extensively investigated [35,36]. Further research is needed to assess the potential impact of co-administration with newly developed vaccinations, considering the aforementioned factors, in order to inform policy decisions and optimize vaccination strategies.

Future research should continue to explore the dynamics of vaccine hesitancy, literacy, health systems, and coverage rates, especially in the context of co-administration. Investigating actual adherence to COVID-19 and influenza co-administration in diverse populations would enhance our understanding of strategies and potential barriers. Longitudinal studies examining the impact of policy changes, healthcare system factors, and socioeconomic determinants on vaccination coverage can provide valuable insights. Additionally, qualitative research exploring individual beliefs, attitudes, and experiences related to co-administration and vaccination uptake can shed light on the factors influencing decision-making and inform targeted interventions.

### 4.2. Limitations

Several limitations should be considered when interpreting the findings of this study. Firstly, the comparison of influenza vaccination coverage and COVID-19 vaccination rates was hampered by a discrepancy in age groups. We used influenza vaccination coverage among persons ≥ 65 years of age, while for COVID-19 vaccination rates, we used the general population or coverage among persons ≥ 60 years of age. We expect that COVID-19 vaccination among persons ≥65 years of age might be slightly higher compared to persons ≥60 years of age and moderately higher compared to the general population, as they were more likely to develop severe symptoms and complications [37]. This would mean that the COVID-19 vaccination data might be an underestimation of true coverage among those ≥65 years of age. Secondly, there was a limitation in units being compared. We used seasonal influenza vaccination coverage, while for COVID-19, we used coverage of the primary series, first booster, second booster, or the number of vaccine doses given per 100 citizens. This reflects a variation in data availability and reporting methods. Caution should be exercised when directly comparing these vaccination coverage estimates. Thirdly, policies on co-administration did not always go in sync with influenza season. COVID-19 primary and booster vaccinations were implemented as soon as possible, whereas influenza vaccinations were implemented during regular influenza season, i.e., August until March in the Northern Hemisphere. Uncertainty remains regarding the success of co-administration due to terms of the seasonality of COVID-19 and influenza. Lastly, having a policy on co-administration in place does not mean that it is adhered to. In this study, we focused solely on the policy of co-administration and vaccination coverage and did not incorporate data on adherence, the timing of availability of the vaccines, or information around execution, reimbursements, healthcare systems, vaccination hesitancy, or socioeconomic disparities. It is worth noting that while this study may provide insights into the relationship between co-administration policies and vaccination coverage, it cannot provide a causal analysis or identify specific mechanisms driving the observed associations.

Despite these limitations, this study’s strengths lie in the inclusion of a substantial number of countries and the size of the populations captured, as well as the consideration of multiple influenza and COVID-19 vaccination seasons. This allows for a broader understanding of the overall trends and patterns observed.

## 5. Conclusions

In conclusion, this study contributes to the understanding of policy on co-administration and vaccination coverage for influenza and COVID-19. The observed correlation between high vaccination rates for one vaccine and corresponding rates for the other underscores the importance of a coordinated approach to immunization. Data related to influenza coverage, COVID-19, and especially co-administration should be improved for a more comprehensive understanding of public health strategies. By implementing evidence-based approaches, improving data, and conducting further research, public health authorities can optimize vaccination efforts and minimize the impact of vaccine-preventable diseases on global health.

## Figures and Tables

**Figure 1 vaccines-12-00216-f001:**
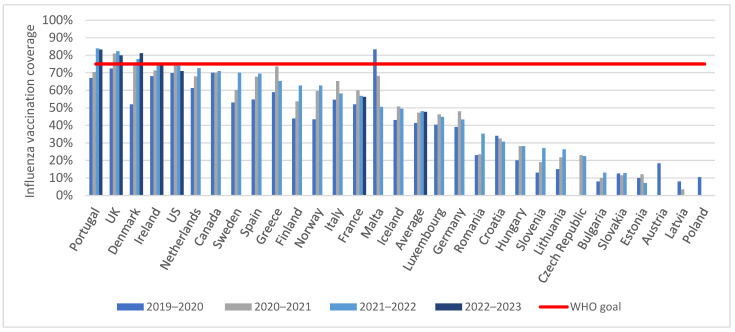
Influenza vaccination coverage among persons ≥ 65 years in the European Union, UK, US, and Canada between the 2019–2020 season and the 2022–2023 season. Countries without data included Belgium and Cyprus.

**Figure 2 vaccines-12-00216-f002:**
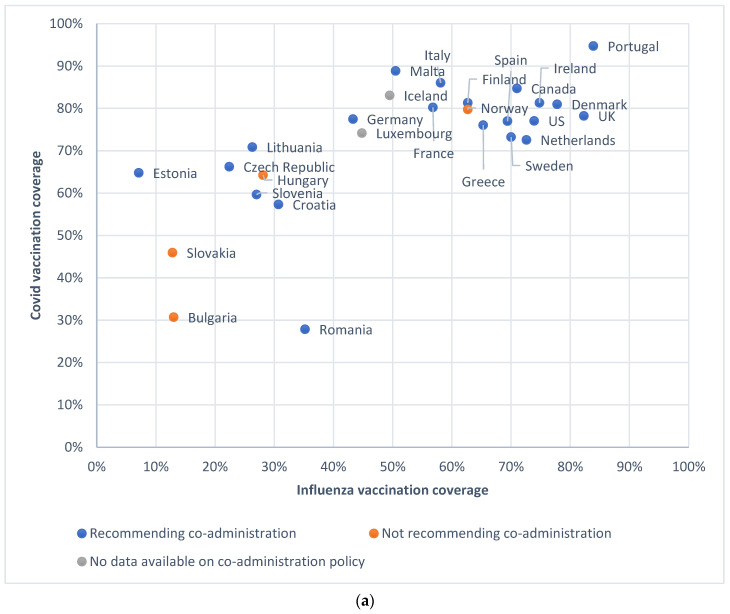
(**a**) COVID-19 vaccination coverage (% of total population receiving ≥1 COVID-19 vaccines) in overall population at March 2022 and influenza vaccination coverage among persons ≥65 for season 2021–2022. Countries missing due to lacking influenza coverage data for season 2021–2022: Austria, Belgium, Cyprus, Latvia, and Poland. Due to lacking data COVID-19 vaccination coverage of Romania at 27 September 2021 was used. (**b**) Number of COVID-19 vaccines given per 100 citizens between August 2021 and March 2022 and influenza vaccination coverage among persons ≥65 years for season 2021–2022. Countries missing due to lacking influenza coverage data for season 2021–2022: Austria, Belgium, Cyprus, Latvia, and Poland. Due to lacking data, only COVID-19 vaccines given per 100 citizens of Hungary were used between 28 September and 21 March. (**c**) COVID-19 vaccination coverage (receiving complete basic series) among persons ≥60 at March 2022 and influenza vaccination coverage among persons ≥65 for season 2021–2022. Countries missing due to lacking influenza coverage data for season 2021–2022: Austria, Belgium, Cyprus, Latvia, and Poland. Due to no data on COVID-19 vaccination in the ECDC COVID-19 Tracker, UK, US, and Canada were removed from this figure.

**Figure 3 vaccines-12-00216-f003:**
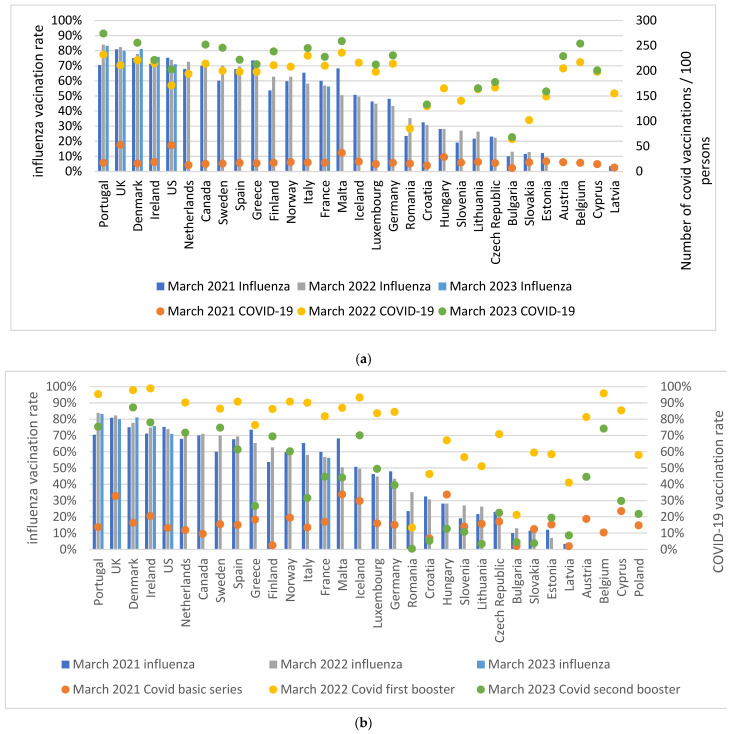
(**a**) Influenza vaccination coverage among persons ≥65 years and number of COVID-19 vaccinations/100 citizens on 31 March 2021, 2022, and 2023. Due to lacking data, we used an earlier date for the number of COVID-19 vaccinations/100 citizens in 2023 for Canada (2 February 2023), Cyprus (22 February 2023), and Hungary (4 December 2022). (**b**) Influenza vaccination coverage among persons ≥65 years and COVID-19 vaccination rate among persons ≥60 years on 31 March 2021 (basic series), 2022 (first booster), and 2023 (second booster).

## Data Availability

Publicly available datasets were analyzed in this study. This data can be found via https://ourworldindata.org accessed on 15 May 2023 and https://qap.ecdc.europa.eu/public/extensions/COVID-19/vaccine-tracker.html#uptake-tab accessed on 21 January 2024. Additional data utilized in this study can be accessed through the references listed in the Appendix A.

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
