# Peer review of "Co-Administration of Influenza and COVID-19 Vaccines: Policy Review and Vaccination Coverage Trends in the European Union, UK, US, and Canada between 2019 and 2023"

_vaccines, 2024, doi:10.3390/vaccines12020216_

Round 1

Reviewer 1 Report

Comments and Suggestions for Authors

"This manuscript, titled 'Co-administration of Influenza and COVID-19 Vaccines: Policy Review and Vaccination Coverage Trends in Europe, UK, US, and Canada between 2019–2023,' aims to analyze the trends and policies regarding the co-administration of these vaccines in the mentioned regions. The manuscript might be more suitable as a short communication or research letter rather than a full paper, as it primarily relies on data collection and presentation. The topic is highly relevant, and the data presented is important; however, the manuscript seems to lack depth in articulating real-world implications, appearing more descriptive than analytical, without drawing specific conclusions from the data.

Here are my recommendations:

It is important to better explain why you are comparing influenza vaccinations in older populations with COVID-19 vaccinations in the general population. This comparison could lead to skewed interpretations due to the differing age demographics and risk profiles. Consider adding a supplementary analysis where you only analyze countries reporting COVID-19 data for those over 65.

I suggest expanding the discussion to include a thorough examination of the potential impact of your findings on public health policies and vaccination strategies, discussing how these findings could influence future policy decisions. E.g., your analysis might benefit from integrating both vaccine literacy/confidence reasoning: specifically, explore on one hand how public perceptions, misinformation, and vaccine literacy may have influenced uptake, depending on coadministration policies. 

Reviewer 2 Report

Comments and Suggestions for Authors

The paper entitled “Co-administration of influenza and COVID-19 vaccines: policy review and vaccination coverage trends in Europe, UK, US and Canada between 2019–2023” describes publicly available data on vaccination coverage in EU/EEA countries, UK, USA and Canada.

The paper does not add much to well known facts about significant differences in vaccination rates within the EU/EEA against infectious diseases for which vaccination is not mandatory.

The title should be changed – authors are not describing influenza and COVID-19 vaccination policy and coverage in Europe but in EU/EEA countries.

Please explain, why “Our World in Data” were used for COVID-19 vaccination coverage not COVID-19 vaccine tracker? COVID-19 vaccine tracker offers more data on vaccination coverage in EU/EEA as countries were requested to report basic indicators (number of vaccine doses distributed by manufacturers, number of first, second, additional and unspecified doses administered) and data by target groups at national level.

Line 125, line 131 and line 138 – Appendix 1, 2 and 3 are mentioned in the text, but are not accessible to the reviewers. The appendices or supplementary material should be added with titles inserted in the end of the main text.

Availability of the data - Due to the heterogeneity of the data sources and thus also the reliability regarding the recommendation of the co-administration of the flu vaccine and the COVID-19 vaccine, it would be appropriate to add the source of the vaccination policy for each country included in the study.

Line 138 –– Data on basic vaccination for 2023 were not available for 6 countries via “Our World in Data” but the data are surely available from other sources (e.g. COVID-19 vaccine tracker). 

Line 156 – The paper contains Figure 2 a and 2b but only Figure 2 is mentioned in the Results section.

Line 167 – The statement “The decrease in COVID-19 vaccinations given per capita at later timepoints may be due to earlier availability of the primary series in the U.S.“  should be moved to Discussion section and compared to e.g. UK COVID-19 vaccine coverage data. UK and USA March 2021 vaccination coverage are comparable but higher number of vaccinations per 100 persons in UK in March 2022 are documented. It might be concluded that early availability had no impact on 2022 COVID-19 vaccination coverage.

Line 174 – Discussion starts in line 174 – one short paragraph only. The discussion of the patterns of differences between countries is scant, with insufficient references. It is necessary to expand the discussion by considering what the possible causes are and specifically how to address them.

Line 181 – There are no statistics tests done to support the correlation between influenza and COVID-19 vaccination coverage - it seems the claim in the sentence relies on visual impression (Figure 2 a and b).

Line 204 – 3.3. Limitations – enumeration is wrong.

Line 232 – 3.4. Interpretation (enumeration is wrong) – I guess that Interpretation is a part of Discussion.  

Reviewer 3 Report

Comments and Suggestions for Authors

We must first thank the authors for their efforts in this interesting research, and the results provide useful information about influenza and COVID-19 vaccinations, including a diversity of countries with different vaccination policies. However, there are major issues that the authors must address to improve research soundness.

The authors are aware and describe the limitations on the information; the effort to integrate different countries must be recognized; however, the heterogeneity of the information, the temporal differences (influenza vs. COVID-19), and segmented by populations, by age, particularly 65 years or older. This generates complex information biases and compromises the interpretation of the results and, therefore, the conclusions.

Limitations:

Age Group Discrepancy: Comparison between influenza vaccination coverage (people aged ≥65 years) and COVID-19 vaccination rates (general population) was hampered by a discrepancy in age groups. This difference affects the comparability of the data.

Variation in Units: Seasonal influenza vaccination coverage was used as the unit of measurement, whereas for COVID-19 vaccination, the number of vaccine doses per 100 citizens or the coverage of the primary series was used. Variations in data availability and reporting methods make direct comparison difficult.

Timing of Vaccination Policies: Policies on vaccine co-administration do not always align with the flu season. Primary and booster vaccinations against COVID-19 were implemented as soon as possible, while influenza vaccines were administered during the regular season, which may have impacted coverage rates.

Data Limitations: The study focused solely on co-administration policy and vaccination coverage and did not incorporate data on adherence, vaccine availability, implementation, reimbursement, health systems, vaccine hesitancy, or socioeconomic disparities, which are factors that could influence coverage.

Causal Analysis: Provides insights into the relationship between co-administration policies and vaccination coverage, but does not offer a causal analysis or identify specific mechanisms driving the observed associations.

Implications for biased interpretations

Age Discrepancy: Can lead to erroneous interpretations of the effectiveness of policies and their real impact on vulnerable groups.

Variation in units: Direct comparison is difficult and could lead to misleading conclusions regarding the relative effectiveness of vaccination campaigns.

Policy Temporality: Can bias the perception of the success of co-administration by not considering the seasonal impact.

Data Limitations: Omission of key factors may result in an incomplete understanding of vaccination coverage, ignoring actual barriers and facilitators.

Causal Analysis: Lack of causal analysis limits the ability to understand and address the reasons behind coverage rates, which can lead to less informed policies and recommendations.

Despite these limitations, the conclusions are not sound, and the research at this point is not recommended for publication in its actual form. We strongly recommend analyzing countries with homogeneous information by consistent age groups (≥ 65 years). The percentage of vaccination as computed must be biased, so an alternative approach should be used. 

Round 2

Reviewer 2 Report

Comments and Suggestions for Authors

No additional comments.

Reviewer 3 Report

Comments and Suggestions for Authors

The authors have responded to the best of their ability to the considerations from the first review. Therefore, I believe that the article provides new results consistent with the literature, identifies limitations, and highlights new gaps in the literature that will be useful to readers. Therefore, I recommend its acceptance for publication.